# Beyond NMDA Receptors: Homeostasis at the Glutamate Tripartite Synapse and Its Contributions to Cognitive Dysfunction in Schizophrenia

**DOI:** 10.3390/ijms23158617

**Published:** 2022-08-03

**Authors:** Hagar Bauminger, Inna Gaisler-Salomon

**Affiliations:** 1School of Psychological Sciences, Department of Psychology, University of Haifa, Haifa 3498838, Israel; hagarbauminger@gmail.com; 2The Integrated Brain and Behavior Research Center (IBBRC), University of Haifa, Haifa 3498838, Israel

**Keywords:** schizophrenia, cognitive deficits, glutamate tripartite synapse, neuron–astrocyte interactions, animal models

## Abstract

Cognitive deficits are core symptoms of schizophrenia but remain poorly addressed by dopamine-based antipsychotic medications. Glutamate abnormalities are implicated in schizophrenia-related cognitive deficits. While the role of the NMDA receptor has been extensively studied, less attention was given to other components that control glutamate homeostasis. Glutamate dynamics at the tripartite synapse include presynaptic and postsynaptic components and are tightly regulated by neuron–astrocyte crosstalk. Here, we delineate the role of glutamate homeostasis at the tripartite synapse in schizophrenia-related cognitive dysfunction. We focus on cognitive domains that can be readily measured in humans and rodents, i.e., working memory, recognition memory, cognitive flexibility, and response inhibition. We describe tasks used to measure cognitive function in these domains in humans and rodents, and the relevance of glutamate alterations in these domains. Next, we delve into glutamate tripartite synaptic components and summarize findings that implicate the relevance of these components to specific cognitive domains. These collective findings indicate that neuron–astrocyte crosstalk at the tripartite synapse is essential for cognition, and that pre- and postsynaptic components play a critical role in maintaining glutamate homeostasis and cognitive well-being. The contribution of these components to cognitive function should be considered in order to better understand the role played by glutamate signaling in cognition and develop efficient pharmacological treatment avenues for schizophrenia treatment-resistant symptoms.

## 1. Introduction

The definition of cognition is broad, encompassing diverse memory, learning and attention capacities as well as higher executive functions. Memory, classified along several axes, e.g., working memory (WM) vs. long-term memory and declarative vs. implicit memory [1], was argued to provide the infrastructure for other cognitive capacities [2]. These include cognitive flexibility, i.e., the ability to adjust to changing demands and circumstances, and response inhibition, i.e., the ability to control behavior in order to choose an adaptable reaction and suppress unwanted responses. These interrelated cognitive domains set the foundations for higher executive functions, e.g., decision making, problem solving, and executive control [3].

Aberrant cognition is a transdiagnostic feature of several psychiatric illnesses, including schizophrenia (SZ), depression and anxiety disorders [4]. In SZ, cognitive deficits commonly appear several years before the onset of psychotic symptoms along with the negative symptoms’ emergence, are detected in first-degree relatives, high-risk individuals, and first-episode patients, and reliably predict functional outcomes later in life [5].

Despite their centrality in disease prediction and progression, the underlying neurobiological basis of cognitive dysfunction in SZ remains poorly understood. Moreover, the ability of currently prescribed dopamine-based SZ drugs to address cognitive impairments is limited [6]. Cognitive deficits were suggested to be associated with glutamate abnormalities in fronto-temporal brain regions [7]. In this review, we propose the glutamate tripartite synapse as a putative target of new treatment options for cognitive dysfunction in SZ. We focus specifically on WM, recognition memory, cognitive flexibility, and response inhibition. These deficits are part of the executive control network that is disrupted in SZ [8], can be readily measured in humans and animals, and rely on well-delineated cross-species neural circuitry. We outline commonly used tasks for measuring these cognitive domains in humans and animals and summarize data on brain regions and circuits implicated in these domains. We then discuss the components of the glutamatergic tripartite synapse and their interactions, presenting evidence from human and animal studies for their role in cognition. We argue that disrupted homeostasis at the tripartite synapse is significantly involved in impaired cognitive function, and that restored glutamate homeostasis should be considered as a marker of therapeutic efficacy and a target for new drug development.

## 2. Measuring Cognition in Humans and Rodents: Tasks and Circuitry

WM, recognition memory, cognitive flexibility, and response inhibition are translational measures of cognitive well-being with shared neural circuitry (Table 1).

### 2.1. Working Memory

Analogous to a mental sketchpad, WM is an active form of memory that retains information for a short period of time, usually minutes. In humans, WM is measured using, e.g., the Wechsler Memory Scale-III (WMS-III:33) digit and spatial span subtests [8] or the spatial span test in the Cambridge Neuropsychological Test Automated Battery (CANTAB) [9]. In animals, WM is assessed in T- or Y-maze choice alternation tasks, which rely on an animal’s natural propensity to select a previously unexplored arm as a hallmark of a memory trace of the visited arm [10]. Delayed match/non-match-to-sample (D(N)MS) tasks can also be used in humans and rodents [11,12]. Intact function of the dorsolateral prefrontal cortex (DLPFC), homologous to the rodent medial PFC (mPFC) [13], and PFC connections with the hippocampus and mediodorsal thalamus (MDT), are critical for WM function [14,15,16].

### 2.2. Recognition Memory

Recognition memory relies on episodic memory, defined as a ‘memory for personally experienced events’ [17]. Recognition tasks require intact memory of a familiar stimulus, and the ability to detect and encode information of a novel one; they can be adjusted to detect deficits in different memory domains by varying the retention period, the number of exposures to familiar stimuli and the recognition target. In humans, such tasks include, e.g., the Relational and Item-Specific Encoding (RISE) task [18], the Verbal Recognition Memory (VRM), and Emotion Recognition (ER) tasks in CANTAB [9]. In animals, the Novel Object Recognition (NOR) task and Object Location Task (OLT) are commonly used to examine recognition memory [19]. Social recognition, testing the ability to detect a non-familiar social stimulus, is also commonly used [20]. Depending on the specifics of the task used, novelty recognition relies on intact function of the prelimbic mPFC [21], perirhinal cortex [22], and hippocampal circuity [23].

### 2.3. Cognitive Flexibility

Cognitive flexibility is the ability to appropriately adjust behavior according to changing environmental demands [24], is measured in humans using, e.g., the Wisconsin Card Sorting Test (WCST) and the CANTAB intradimensional/extradimensional attentional-shift test [9]. These tasks contain a ‘reversal’ component, requiring subjects to modify decisions according to changes in reinforcement contingencies, and an attentional set-shifting component, relying on the ability to discriminate between rewarded dimensions [3]. In animals, reversal learning (also termed intradimensional (ID) shifting) and attentional (extradimensional) set-shifting (EDSS) can be assessed using diverse paradigms and stimuli. For example, in the Birrell–Brown task, animals are trained to discriminate between two bowls, differing in dimensions such as odor and texture, to obtain a food reward. While the ID component requires the animal to reverse its choice within the same dimension, the ED component involves a switch in the discriminating dimension [25]. A similar negative reinforcement-based task requires animals to locate a platform placed in a particular arm in a water T-maze, then reverse the location (ID shift) or the platform-locating rule (spatial vs. visual cue; ED shift) [26]. 

The ability to perform ID/ED shifts may be mediated by different neural circuitry: while ID shifts are facilitated by the orbitofrontal cortex (OFC) and its connections with the ventral and dorsal striatum [27], EDSS is mediated by mPFC-MDT [28] and ventral striatum connections [29]. 

### 2.4. Response Inhibition

Response inhibition is the ability to inhibit inappropriate responses and can be assessed in humans using the AX-Continuous Performance Test (AX-CPT: 35) [30]. Preclinical studies commonly use the stop-signal task (SST), measuring latency to stop the reaction as a proxy of inhibitory processes. Typically, animals are trained to produce a “go” response (e.g., press two levers in fast sequence), and to inhibit a response to the second lever when a stop signal (a tone) appears [31]. Response inhibition can be viewed as an integral component of flexible and adaptive behavior, since it requires the ability to rapidly suppress contextually inappropriate responses. Unsurprisingly, the neural substrates of EDSS and response inhibition are similar. For example, fronto-basal ganglia circuits are activated during response inhibition tasks [32].

**Table 1 ijms-23-08617-t001:** Measuring cognition in humans and rodent models.

Cognitive Capacity	Human Measurement ^#^	Tasks Commonly Used in Rodents	Brain Circuitry
**Memory**			
WM	Digit and spatial span subtests, WMS-III:33 [8]; spatial span test, CANTAB [9]; DNMS, CANTAB [12].	Alternation in a T- or a Y-maze [10]; DNMS tasks [11].	DLPFC/mPFC, PFC connectivity with the hippocampus and MDT [14,15,16].
Episodic memory (visual/visuospatial/social recognition)	RISE task [18]; VRM, ET tasks, CANTAB [9].	NOR, OLT [19] social interaction [20].	Prelimbic mPFC [21], Perirhinal cortex [22], hippocampal circuity [23].
**Cognitive Flexibility**			
Reversal/Attentional Set Shifting	WCST; CANTAB IED [9].	Appetitive: Birrel–Brown task [25]. Aversive: Water T-maze with reversal and attentional set-shifting components [26].	Reversal: OFC-ventral, dorsal striatum [27]. Set-shifting: mPFC—MDT [28], ventral striatum [29].
**Response inhibition**			
Response Inhibition	AX-CPT: 35 [30].	SST [31].	Fronto-basal ganglia circuity [32].

^#^ Tasks listed in this column are representative example of human measurements, reviewed elsewhere [33]. WM = working memory; WMS-III:33 = Wechsler Memory Scale-III; CANTAB = the Cambridge Neuropsychological Test Automated Battery; DNMS = delayed match/non-match-to-sample; DLPFC = dorsolateral prefrontal cortex; mPFC = medial prefrontal cortex; PFC = prefrontal cortex; MDT = medial dorsal thalamus; RISE = Relational and Item-Specific Encoding; VRM = Verbal Recognition Memory; ET = Emotion Recognition; ACC = anterior cingulate cortex; WCST = the Wisconsin Card Sorting Test; IED = interdimensional/extradimensional; OFC = orbitofrontal cortex; AX-CPT: 35 = the AX-Continuous Performance Test; SST = stop-signal task.

## 3. Cognitive Dysfunction and Glutamate in SZ

Several lines of evidence support the relevance of abnormal glutamate transmission to cognitive dysfunction in SZ. First, glutamate perturbations are found in the prodromal phase of SZ [34], which is also the developmental time period in which the cognitive deficits of the disease commonly emerge [35]. Second, proton magnetic resonance spectroscopy (1H-MRS) studies show that glutamate abnormalities in cortico-thalamic-hippocampal circuits are associated with cognitive deficits in SZ [7]. Third, genomic investigations link aberrant cognition to disrupted glutamate in SZ: a large-scale analysis of genome-wide association studies (GWASs) identified 100+ loci linked to both general cognitive function and SZ [36], and GWAS studies indicate that genetic variants related to cognitive trait impairment in SZ partake in the glutamate receptor activity network [37]. 

Despite the progress these studies have allowed in cognition research, the relationship between glutamate abnormalities and cognitive dysfunction remains to be clarified. It is unclear, for example, whether enhanced and/or reduced glutamate release results in pathological cognitive decline. In order to thoroughly understand this relationship, it is mandatory to examine the different components required for homeostasis at the glutamate tripartite synapse. 

## 4. The Glutamate Tripartite Synapse: Involvement in Cognitive Capacities

Glutamatergic neurotransmission occurs within the confines of the tripartite synapse, which consists of a presynaptic terminal, a post-synaptic spine, and an astrocyte [38]. Glutamate plays a critical role in neurotransmission as well as in energy supply. As detailed below and demonstrated in Figure 1, glutamate can be derived from α-ketoglutarate or recycled via the astrocyte–neuronal glutamate–glutamine cycle, where glutamine is hydrolyzed to glutamate by the neuronal rate-limiting enzyme glutaminase (GLS1) [39]. Glutamate is packed into synaptic vesicles by vesicular glutamate transporters (vGluts) [40]. After release, glutamate acts upon ionotropic (NMDA, AMPA, kainate) and metabotropic receptors (mGluRs) located on pre- and postsynaptic membranes, as well as on astrocytes [41]. Excess glutamate is cleared from the extracellular space by neuronal and astrocytic excitatory amino acid transporters (EAAT1-5). In astrocytes, glutamate is converted by glutamine synthetase (GS) to glutamine, which is released from astrocytes and taken up by glutamatergic neurons via sodium-coupled neutral amino acid transporters (SNATs) [42]. Glutamate is also released into the extra-synaptic space by astrocytes via the cystine–glutamate antiporter (Xc-), an amino acid transporter that exchanges glutamate for cystine [43]. While in this review we focus on the components of the glutamate tripartite synapse, clearly the interaction of glutamate with other neurotransmitter systems, e.g., DA and GABA, plays an important role in cognition. 

Below, we consider the contribution of different glutamate tripartite synapse components to WM, recognition memory, cognitive flexibility and response inhibition, and summarize the interactions between different tripartite synapse components, with special emphasis on neuron–astrocyte crosstalk (Table 2). 

**Figure 1 ijms-23-08617-f001:**
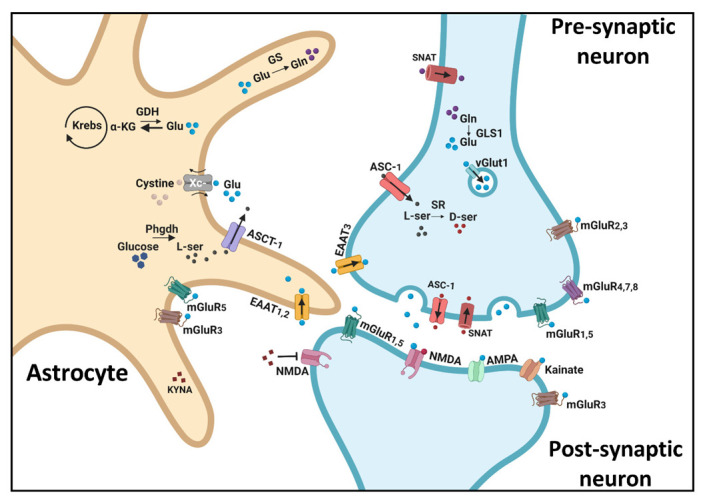
The glutamate tripartite synapse. Neuronal and astrocytic processes involved in controlling glutamate homeostasis. a-KG = α-ketoglutarate; AMPA = α-amino-3-hydroxy-5-methyl-4-isoxazolepropionic acid; Asc-1 = alanine–serine–cysteine transporter 1; ASCT-1 = alanine–serine–cysteine–threonine transporter 1; D-ser = D-serine; EAAT = excitatory amino acid transporters; GDH = glutamate dehydrogenase; Gln = glutamine; GLS1 = glutaminase; Glu = glutamate; GS = glutamine synthetase; KYNA = kynurenic acid; L-ser = L-serine; mGluR = metabotropic glutamate receptors; NMDA = N-methyl-D-aspartate; Phgdh = 3-phosphoglycerate dehydrogenase; SNAT = sodium-coupled neutral amino acid transporters; SR = serine racemase (also reported to be found in astrocytes and post-synaptic neurons, see text); vGluts = vesicular glutamate transporter 1; Xc- = cystine–glutamate antiporter.

### 4.1. The NMDA Receptor 

The NMDA receptor (NMDAR) is a heteromeric assembly composed of two obligatory GluN1 subunits and different combinations of the GluN2 and/or GluN3 subunits. Activation of this voltage-dependent channel necessitates membrane depolarization and binding of both glutamate and a co-agonist (glycine or D-serine) to separate binding sites. Receptor activity can be blocked by kynurenic acid (KYNA) a metabolite of the kynurenine pathway which is synthesized and released by astrocytes [44] and binds to NMDAR at the glycine site [45]. The NMDAR can also be blocked by exogenous non-competitive antagonists, e.g., phencyclidine (PCP), ketamine, or MK-801 (dizocilpine), which bind to the PCP site within the channel pore [46]. 

NMDAR function relies heavily on neuron–astrocyte crosstalk. A prominent example of this is the provision of D-serine via the glia–neuron serine shuttle. L-serine is produced in astrocytes from glucose by the astrocytic enzyme 3-phosphoglycerate dehydrogenase (Phgdh) [47], released by neutral amino acid exchangers (ASCT-1; alanine–serine–cysteine–threonine transporter 1), and taken up by neurons via neutral amino acid antiporters such as alanine–serine–cysteine transporter 1 (Asc-1) [48,49]. D-serine is synthesized from L-serine by the enzyme serine racemase (SR). Some controversy remains regarding the synaptic synthesis site of D-serine [50]. While initially D-serine synthesis by SR was shown to occur predominantly in astrocytes [51], others have shown that SR is responsible for presynaptic neuronal D-serine synthesis [52], which is then released from neurons via Asc-1 [53]. Recent evidence suggests postsynaptic localization of SR in the dendrites of CA1 pyramidal neurons, where D-serine regulates NMDAR activity in an autocrine fashion [54]. D-serine clearance is achieved via system A transporters [55] and then degraded by d-Amino Acid Oxidase (DAO), which is regulated by D-Amino Acid Oxidase Activator (DAOA; G72) [56].

NMDAR activation leads to Ca^2+^ influx, ultimately facilitating long-term potentiation, a cellular plasticity phenomenon that underlies systemic learning and memory processes [57]. The role of the NMDAR in cognition has been extensively reviewed elsewhere [58]; here, we will briefly summarize findings relevant to specific cognitive capacities, and will describe the interactions between NMDAR and other tripartite synapse components. 

Non-competitive NMDAR blockade leads to a state that resembles the full spectrum of SZ symptoms, including cognitive dysfunction, in healthy individuals [59] and exacerbates cognitive symptoms in patients with SZ [60], hallmark findings that sparked interest in the involvement of glutamate in SZ-related cognitive dysfunction. For example, sub-anesthetic doses of ketamine in healthy volunteers induces WM impairments and attention deficits [61], and reproduces deficits in other cognitive domains [59]. Additionally, increased KYNA is found in PFC of SZ patients and is linked to cognitive abnormalities, e.g., spatial WM deficits, in preclinical rat models of SZ [45]. In further support of the relevance of NMDAR dysfunction to cognitive abnormalities, genetic variation in the NMDAR subunit *GRIN2A* is a consistent finding in SZ patients [62,63] and GluN2B (*GRIN2B*) subunit genetic variation contributes to variability in cognitive function in SZ [64].

In preclinical research, pharmacological blockade of the PCP site impairs WM [65], NOR, reversal learning, and EDSS [66], as well as response inhibition [67,68]. Genetically induced NMDAR malfunction, generally via GluN1 knockdown (KD) or deletion, or via reduced co-agonist affinity or availability, similarly leads to memory [69,70,71,72], EDSS [73], and response inhibition [74] deficits. Conversely, mice with a genetically induced enhancement of glycine or D-serine availability, e.g., DAO knockout (KO) mice or mice with a forebrain pyramidal neuron-specific deletion of the glycine transporter, show procognitive abilities such as facilitated spatial learning [75], improved reversal learning [76], enhanced NOR and OLT performance [77]. These studies support the importance of neuronal–astrocytic crosstalk for proper NMDAR function, and provide evidence for the putative effectiveness of glycine site activation in treating cognitive abnormalities; human studies with glycine site agonists have been inconclusive and difficult to interpret, possibly due to NMDAR saturation and/or placebo effects [78,79]. 

A major insight emerging from animal studies is that NMDAR blockade results in excess pyramidal neuron firing, leading to surplus glutamate release in fronto-hippocampal regions [80,81]. Accordingly, NMDAR blockade in adolescence alters vGlut1 hippocampal protein [82] and prelimbic mPFC mRNA expression [83], and leads to abnormally high glutamate levels [84]. Several mechanisms have been proposed to account for this finding, including reduced GABA activity subsequent to hypoactivity of NMDA receptors on parvalbumin (PV) neurons [85]. However, other findings indicate that PV interneurons do not play a critical role in this process [86], suggesting that other mechanisms may account for NMDAR blockade-induced excess glutamate release and the ensuing behavioral abnormalities.

One such mechanism could be reduced glutamate clearance by astrocytic EAATs. Subchronic ketamine administration to adult [87] and adolescent mice [88] reduces the expression of EAAT2, resulting in enhanced extracellular glutamate concentration. Deficient glutamate reuptake by astrocytic transporters results in glutamate spillover leading to loss of input specificity, extra-synaptic receptors activation and possibly excitotoxicity-induced cell death mechanisms [89]. Thus, NMDAR hypofunction can lead to glutamate spillover by modifying processes within the glutamate tripartite synapse, independently of the interaction between glutamate transmission with other neurotransmitter systems. 

### 4.2. AMPA and Kainate Receptors

α-amino-3-hydroxy-5-methyl-4-isoxazoleproprionic acid receptors (AMPAR) and kainate receptors are involved in fast synaptic transmission; glutamate binding to these receptors results in a conformational change and cation influx. The core of the AMPAR complex is a hetero-tetrameric ion channel consisting of combinations of four different subunits (GluA1–4) [90]. Kainate receptors are homo- or hetero-tetrameric glutamate gated ion channels composed of five subunits, GluK1-5. Sodium and calcium permeability at AMPAR and kainate receptors is affected by subunit assembly, as well as RNA splicing and editing [91,92]. Astrocyte-dependent processes impact subunit composition, synaptic distribution, and functionality; for example, astrocyte-secreted components control the expression of both GluA1 and GluA2 in developing glutamate neurons [93,94]. GluK2-containing receptors are found on astrocytes, where they participate in regulating neuronal glutamate release [95].

Abnormalities in AMPA and kainate receptors are implicated in SZ psychopathology. For example, in SZ patients genetic variation is found in the AMPAR subunits *GRIA1* [96] and *GRIA3* [63]. Additionally, abnormal expression of AMPA and kainate receptor subunits was found postmortem in the PFC of patients with SZ [97].

Preclinical studies support the relevance of AMPA and kainate receptors to cognitive dysfunction. *Gria1^−/−^* mice display spatial WM deficits [98], attributed to AMPAR abnormalities in forebrain pyramidal neurons [99]. Pharmacological blockade of AMPA and kainate receptors in the posterior parietal cortex impairs accuracy in a touchscreen-based visuospatial WM task [100], and infusion of the kainate receptor antagonist UBP-302 to the perirhinal cortex impairs NOR [101]. In addition, GluR-A*^−/−^* mice show deficits in reversal learning in an appetitive elevated plus-maze task [102] and GluK2 KO mice show reversal learning deficits in the water maze [103]. AMPAR blockade in rat mPFC disrupts set-shifting, but not reversal [104], and impairs performance in the SST response inhibition task [105]. 

Conversely, compounds that potentiate AMPAR activity enhance cognitive performance and reverse NMDAR blockade-induced cognitive deficits. In healthy humans, sub-anesthetic doses of ketamine-induced verbal memory deficits are ameliorated by an AMPAR potentiator [106]. In rats, positive modulation of the AMPAR reverses PCP-induced NOR deficits [107]. Given on their own, ampakines enhance spatial WM performance [108], a procognitive effect that has been attributed to their ability to enhance synaptic efficacy and lead to efficient astrocytic glucose utilization [109]. Together, these findings suggest that non-NMDA ionotropic glutamate receptors contribute to fundamental aspects of cognition, and that astrocyte–neuron interactions play a role in mediating their effects.

### 4.3. Metabotropic Glutamate Receptors (mGluRs)

mGluRs are a family of G-protein coupled receptors. The 8 mGluR subtypes (mGlu1-8) are classified into three main groups (I-III). Group I receptors (mGluR1,5) are predominantly expressed on post-synaptic neurons and regulate NMDAR activity but are also found presynaptically. mGluR5 is also detected on astrocytes [41,110]. Group II receptors (mGluR2,3) are mainly expressed on presynaptic neurons outside the synaptic active zones; mGluR3 is also expressed on post-synaptic neurons and astrocytes. Group III receptors (mGluR4,6,7,8) are predominantly expressed on presynaptic neurons in synaptic active zones [111]. While activation of Group II and III mGluRs reduces glutamate release [111], stimulation of Group I presynaptic receptors can either facilitate or inhibit glutamate release [112]. 

Genetic variations in mGluR-encoding genes are associated with SZ, cognitive dysfunction, and dysregulated glutamate signaling. For example, a single-nucleotide polymorphism (SNP) in the mGluR3 encoding gene (*GRM3*) is associated with increased risk for SZ, reduced cognitive performance in verbal fluency and verbal list learning tasks, as well as reduced EAAT2 mRNA expression in PFC [113]. Conversely, pretreatment with the mGluR2,3 agonist LY354740 reversed WM deficits induced by sub-anesthetic doses of ketamine, but not learning or attention deficits in healthy volunteers [114]. Although preliminary clinical trials showed mGluR2,3 agonists are effective in ameliorating positive and negative SZ symptoms [115], their effectiveness against cognitive deficits is debatable, and larger clinical trials failed to reproduce their beneficial effects [116].

Preclinical studies show that altering mGluR-mediated transmission affects cognition. Mice with a genetically-induced reduction in mGluR4 display deficits in spatial reversal learning [117]. mGluR7, the only mGluR expressed in CA1 [118], plays a critical role in WM [119].

As in humans, stimulation of mGluRs in rodents was shown to have beneficial effects on cognition, although findings are often inconsistent. For example, LY354740 and LY379268, targeting both mGluR2 and mGluR3, reverse PCP and MK-801-induced WM deficits in a discrete-trial delayed alternation task [120], and NOR [121]. However, the mGluR2,3 agonist LY379268 exacerbates PCP-induced response inhibition deficits in the 5-choice serial reaction time task [122], and is effective in reversing PCP-induced NOR deficits only when co-administered with antipsychotic medications [123]. Interestingly, the specific mGluR2 positive allosteric modulator (PAM) LY487379 enhances EDSS in rats [124]. Another mGluR2 PAM, SAR218645, restores MK-801-induced Y-maze WM deficits in NR1^neo−/−^ mice and NOR deficits in rats [125]. Similarly, the mGluR5 PAM CDPPB reverses MK-801-induced deficits in a four-arm EDSS task [126] and PCP-induced deficits in NOR [127], as does the mGluR4 selective agonist LSP4-2022 [128]. Thus, while the efficacy of pharmacotherapies that activate mGluRs in a non-specific manner is debatable, specific activation of mGluR2,4 and 5 may have procognitive effects, especially under NMDAR hypofunction conditions.

The procognitive abilities of mGluR modulators may be related to their capacity to reduce glutamate release from pyramidal neurons [129]. For example, the mGluR1 PAM VU6004909 was found to reverse WM deficits in spontaneous alternation in the Y-maze following MK-801 administration due to increased inhibitory transmission in the prelimbic mPFC, leading to regulated pyramidal cells activity [130]. The procognitive effects may also involve astrocyte–neuron interactions: glutamate released by astrocytes activates group I mGluRs on presynaptic neurons, resulting in enhanced neuronal glutamate release [131]. Activated astrocytes also promote the expression of new mGluR2/3 and 5 [132].

### 4.4. Glutamate Synthesis and Release Mechanisms

A major source of glutamate is the glutamate–glutamine recycling pathway [39]. Several studies have found reduced GS [133] and increased GLS1 expression in the PFC of patients with SZ, but results are inconclusive and do not associate GS or GLS1 abnormalities with a specific symptom domain [134]. In mice, the GS inhibitor methionine sulfoximine (MSO) impairs OLT when administered in early development, and reduces synaptogenesis in adulthood [135]. Conversely, we have shown that mice with a global deletion of GLS1 are resilient to the PFC-activating effect of ketamine [136] and to deficits in social recognition induced by neonatal MK-801 (in prep). The contrasting effects of GS or GLS1 inhibition on behavior may not be attributed simply to opposite effects on glutamate release since inhibition of both molecules reduces glutamate signaling in the hippocampus [135,136].

vGluts, which pack glutamate into synaptic vesicles, play a critical role in glutamate release. Of the different vGlut isoforms (vGlut1-3), vGlut1, predominantly expressed in the cortex, CA1-CA3 and dentate gyrus [40], may be particularly relevant to cognitive dysfunction in SZ. Postmortem examinations of patients with SZ show area-specific changes in mRNA and protein expression of vGlut1, with inconsistent findings: for example, decreased *vglut1* mRNA levels in DLPFC and hippocampus were found [137], but others have shown no changes in *vglut1* mRNA in DLPFC of SZ patients [138]. Increased *vglut1* mRNA but reduced protein levels have been shown in ACC of patients [139]. vGlut1-deficient mice (vGlut1^+/−^), with reduced glutamatergic neurotransmission, show impaired NOR [140], WM and social memory [141], reversal learning [142,143], and response inhibition [144]. vGlut1 depletion in dorsal hippocampus leads to NOR deficits [145].

Another factor contributing to glutamate dynamics is glutamate dehydrogenase (GDH; gene name: *Glud1*), an enzyme catalyzing the breakdown of glutamate to α-ketoglutarate, followed by further oxidation in the TCA cycle [42]. GDH is mainly expressed in astrocytes, with significantly lower expression levels in neurons, microglia, and oligodendrocytes. Under certain physiological conditions, GDH can also aminate α-ketoglutarate to glutamate [146]. Human studies indicate that GDH enzymatic activity in PFC is increased and mRNA expression levels are reduced in CA1 of patients with SZ [147,148].

Nestin-Cre+;*Glud1^−/−^* mice, with a CNS-specific GDH deletion, show enhanced pyramidal neuron activity, elevated hippocampal glutamate levels, and increased expression of astrocytic glutamate transporters as well as dysregulated NMDA and AMPA receptor subunits expression [148]. Our lab has shown that Nestin-Cre+;*Glud1^−/−^* mice display deficits in NOR, OLT, social recognition, reversal learning, and EDSS [148,149].

While some of these findings associate glutamate elevations with cognitive deficits, manipulations that reduce glutamate release from astrocytes were also found to induce cognitive abnormalities. For example, temporally controlled and astrocyte-specific expression of the tetanus neurotoxin (TeNT), which inhibits vesicle fusion, blocks in vitro glutamate release from astrocytes and induces NOR deficits in mice [150]. Similarly, transgenic astrocyte-specific dominant-negative SNARE (hGFAP-dnSNARE) mice, with reduced glutamate release from astrocytes [151] and disrupted dorsal hippocampus-mPFC synchronization, show WM deficits in a water maze and deficits in NOR which are reversed by D-serine treatment [152].

In sum, manipulations that interfere with glutamate release, either increasing or decreasing transmission via neuronal or astrocytic packaging and release mechanisms, lead to cognitive deficits.

### 4.5. Glutamate Reuptake

At the tripartite synapse, glutamate clearance from the synaptic cleft is a critical process for protecting neurons from glutamate spillover which can lead to excitotoxicity. This process is achieved mostly by the glial transporters EAAT1 and EAAT2 (also known as GLAST and GLT-1, respectively), predominantly expressed on astrocytes [110,153]. Postmortem findings of EAAT’s expression in SZ are inconsistent. Some findings indicate that EAAT2 mRNA and protein are increased in the DLPFC in patients [154], but others report decreased PFC mRNA levels [155]. In addition, decreased protein levels of EAAT1 and EAAT2 were found in the superior temporal gyrus as well as decreased EAAT2 protein levels in the hippocampus [156]. These inconsistencies might be due to area or cell-specific dysregulations of EAATs expression in patients, for a review see [89]. Moreover, altered activity of EAATs is related to cognitive deficits in SZ, for example, worse WCST performance was found among SZ patients that are carriers of EAAT1 and 2 SNPs which leads to lower transporter expression and activity [157].

Rodent studies show that EAAT1-deficient mice (EAAT1^−/−^) display deficits in learning and recognition memory tasks, particularly social recognition [158]. Pharmacological blockade of EAAT1 similarly impairs NOR in mice [159], while enhancement of EAAT1 activity reduces glutamate excitotoxicity in culture [160], providing a possible mechanism for EAAT1-targeting drugs. However, EAAT^+/−^/- mice are resilient to the effects of PCP on NOR [161], in further support of the notion that a delicate balance in glutamate levels at the tripartite synapse must be maintained for intact cognitive function.

EAAT2 malfunction is also associated with cognitive deficits. For example, pharmacological EAAT2 blockade using dihydrokainic acid (DHK) leads to spatial learning and short-term memory deficits [162]. Additionally, pharmacological compounds or genetic manipulations that increase reuptake, e.g., ceftriaxone or riluzole, improve performance in NOR and spatial WM in Alzheimer disease mouse models [163,164]. These findings imply that genetic or pharmacological manipulations of EAATs may hold promise as therapeutic venues targeting cognitive deficits in SZ.

### 4.6. Indirect Astrocytic Effects on Glutamate Dynamics

Neuron–astrocyte dynamics affecting glutamate homeostasis can be regulated indirectly. For example, astrocyte-specific deletion of adenosine A2A receptors leads to altered GLT-1 activity, increased presynaptic glutamate release, NMDAR expression abnormalities, and increased internalization of AMPAR, as well as WM alterations in the Y-maze and radial arm maze [165]. Notably, this behavioral phenotype was prevented by selective GLT-1 inhibition or GluA1,2 endocytosis blockade [166]. Furthermore, manipulations that lead to plastic changes in astrocytes, e.g., enhanced ensheathment of synapses, result in elevated astrocytic glutamate uptake and reduced hippocampal-dependent contextual fear memory [167]. The role of astrocytic plastic changes in other cognitive capacities remains to be determined. 

**Table 2 ijms-23-08617-t002:** Preclinical findings of tripartite synapse components involvement in cognitive function.

Tripartite Synapse Component	Memory		Cognitive Flexibility		Response Inhibition	Effects on Other Components of the Tripartite Synapse
	WM	Recognition Memory	Reversal Learning	Attentional Set-Shifting		
NMDAR antagonists (PCP/ketamine/MK-801)	Radial arm maze, DNMS, delayed alternation task deficits [65].	NOR deficits [66].	Operant task deficits [66].	EDSS deficits in ASST [66].	5CSRTT deficits [67,68].	Ketamine, MK-801: increased mPFC glutamate release [80,81]. Ketamine: reduced hippocampal GLT-1 expression [87,88]. Adolescence MK-801: increased PrL Vglut1 mRNA expression [83]; increased hippocampal Vglut1 protein expression [82].
Genetically induced NMDAR hypofunction	NR1-KD mice: Y-maze deficits [69].			NR1-KD mice: radial arm maze perseverative errors [73].	mPFC/dCA3 NR1 deletion: 5CSRTT deficits [74].	
Glycine site manipulations		Srr^Y269^ mice: OLT deficits; D-serine reversal [71]. GlyT1-KO mice: improved NOR and OLT [77]. DAO^−/−^ mice: improved MWM [75]. SR^−/−^ mice: impaired memory of order of events in object recognition and odor sequence tests [70].	Dao1^G181R^ mice: improved reversal learning in the MWM [76].			Srr^Y269^ mice: reduced D-serine levels [71].
AMPAR	Gria^−/−^ mice: spatial WM deficits in the T and Y maze [98]. CNQX infusion to PPC: TUNL impairment [100]. Ampakine CX516: procognitive influence in a DNMS task [108].	Ampakines (CX546/CX516): reversal of PCP-induced NOR deficits [107].	GluR-A^−/−^ mice: appetitive elevated plus-maze task, impairment [102].	NBQX mPFC injection: EDSS deficits in the Birrell and Brown ASST [104].	NASPM PFC infusion: SST impairment [105].	GluA1flox/flox^CamKCreER^ mice: GluA2 redistribution [99].
Kainate receptors	CNQX infusion to PPC: TUNL impairment [100].	UBP-302 perirhinal cortex infusion: NOR impairment [101].	GluK2 KO: MWM deficits [103].			
mGluRs	mGluR7 KO: 4/8-arm maze task impairment [119]. LY354740: reversal of PCP-induced deficits in the discrete-trial delayed alternation task [120]. SAR218645 NR1^neo−/−^ mice: reversal of Y-maze deficits [125]. VU6004909 pretreatment to MK-801, mice: reversal of MK-801-induced Y-maze deficits [130].	LY379268: reversal of MK-801-induced NOR deficits [121]. LY379268 co-administration with clozapine: reversal of PCP-induced NOR deficits [123]. SAR218645: reversal of MK-801-induced NOR deficits in rats [125]. CDPPB: reversal of PCP-induced NOR deficits [127]. LSP4-2022: reversal of MK-801-induced NOR deficits [128].	mGluR4 KO mice: MWM impairment [117].	LY487379: procognitive effect in the Birrell and Brown ASST [124]. CDPPB: reversal of MK-801-induced EDSS deficits in 4 arm maze ASST [126].	LY379268: exacerbation of PCP-induced 5CSRTT deficits [122].	Astrocytic activation: increased mGluR2,3 and 5 expression [132]. VU6004909 pretreatment to MK-801, rats: reversal of MK-801-induced cortical hyperactivity [130].
Glutamate synthesis and release mechanisms	vGlut1^+/−^ mice: T-maze impairment [141]. dnSNARE mice: MWM deficits [152].	MSO, mice: OLT impairment [135]. vGlut1^+/−^ mice: NOR impairment [140]; social recognition impairment [141]. dorsal hippocampus vGlut1 depletion, mice: NOR impairment [145]. Nestin-Cre+;*Glud1^−/−^* mice: NOR, OLT and social recognition deficits [149]. TeNT astrocytic expression, mice: NOR impairment [150]. dnSNARE mice: NOR deficits [152].	vGlut1^+/−^ mice: MWM deficits [142]; visual discrimination task deficits [143]. Nestin-Cre+;*Glud1^−/−^* mice: IDSS deficits in the water T-maze [148].	Nestin-Cre+;*Glud1^−/−^* mice: EDSS deficits in the water T-maze [148].	vGlut1^+/−^ mice: spatial extinction learning deficits [144].	MSO, mice: decreases CA3 sEPSC, reduced functional synapses and decreased glutamatergic neurotransmission [135]. GLS1 het’ mice: resiliency to ketamine-induced PFC activation; reduced PFC and hippocampal glutamate levels; increased glutamine levels [136]. *vGlut1*^+/−^ mice: reduced hippocampal glutamate levels [168].Nestin-Cre+;*Glud1^−/−^* mice: enhanced pyramidal neuron activity, hippocampal glutamate levels, astrocytic glutamate transporters and NMDA and AMPA receptor subunits expression [148]. hGFAP-dnSNARE mice: reduced glutamate release from astrocytes [151].
Glutamate reuptake	Ceftriaxone, APP/PS1 mice: reversal of MWM deficits [164].Transgenic/ pharmacological EAAT2 restoration, APP_Sw,Ind_ mice: reversal of Y-maze impairments [163].	EAAT1*^−/−^* mice: social recognition impairment [158]. DHK, mice: NOR deficits [159]. EAAT^+/−^ mice: resilience to PCP-induced NOR deficits [161]. DHK infusion, rats: MWM impairment [162]. Transgenic/ pharmacological EAAT2 restoration, APP_Sw,Ind_ mice: reversal of NOR deficits [163].				In vitro ceftriaxone, cultured neurons and astrocytes: increased glutamate reuptake [160]. Ceftriaxone, APP/PS1 mice: upregulated GS activity [164].
Indirect astrocytic effects on glutamate dynamics	Gfa2-A2AR KO mice: Y-maze and radial arm maze deficits; reversal of deficits by DHK or GluA1,2 endocytosis blockade [165].					Gfa2-A2AR KO mice: altered GLT-1 activity, increased glutamate release, NMDAR expression abnormalities and increased AMPAR internalization [165].

WM = working memory; PCP = phencyclidine; MK-801 = dizocilpine; DNMS = delayed non-match-to-sample; NOR = novel object recognition; EDSS = extradimensional set-shifting; ASST = attentional set-shifting task; 5CSRTT = 5 choice serial reaction time task; mPFC = medial prefrontal cortex; GLT-1 = glutamate transporter 1; PrL = prelimbic; Vglut1 = vesicular glutamate transporter 1; dCA3 = dorsal CA3; OLT = object location task; MWM = Morris water maze; PPC = posterior parietal cortex; TUNL = Trial-Unique Non-matching-to-Location; SST = stop-signal task; mGluR = metabotropic glutamate receptors; IDSS = interdimensional set-shifting; EAAT = excitatory amino acid transporters.

## 5. Summary

In this review, we discussed abnormal glutamate transmission at the tripartite synapse as a neural basis for the cognitive dysfunction in SZ and described animal tasks used for measuring WM, recognition memory, cognitive flexibility, and response inhibition in rodent models of psychopathology. We outlined evidence supporting the relevance of aberrant glutamate transmission to cognitive deficits in SZ human research and detailed the neuronal and astrocytic components of the tripartite synapse that contribute to glutamate homeostasis. This summary of findings indicates that excess glutamate at the tripartite synapse leads to cognitive deficits; however, some findings also point to a hypoglutamatergic state as a driver of cognitive dysfunction, implying that deviating from a homeostatic state at this synapse may be detrimental to learning, memory, and attention and is likely to contribute to cognitive decline in SZ and related disorders. While glutamate receptors, and particularly the NMDAR, have been the focus of both preclinical and clinical investigations pertaining to cognition in SZ, future efforts to better understand the neural basis of cognition and drug design for cognitive dysfunction should focus on mechanisms contributing to neuron–astrocyte interactions and the different components that participate in regulating glutamate synthesis, release, and reuptake at the tripartite synapse.

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
