# Peer review of "Beyond NMDA Receptors: Homeostasis at the Glutamate Tripartite Synapse and Its Contributions to Cognitive Dysfunction in Schizophrenia"

_ijms, 2022, doi:10.3390/ijms23158617_

Round 1

Reviewer 1 Report

This good paper which comprehensively discuss the contribution of glutamate synapse in cognitive dysfunction in schizophrenia. The paper is well described and organized and in proper way gather the existing knowledge.

Author Response

Thank you very much for this positive review

Reviewer 2 Report

The manuscript "Beyond NMDA receptors: homeostasis at the glutamate tripartite synapse and its contributions to cognitive dysfunction in schizophrenia" by Bauminger and Gaisler-Salomon summarizes the current knowledge about the involvement of tripartite glutamate synapse in cognitive dysfunctions observed in schizophrenia. The topic is important because currently developed therapies for schizophrenia have only antipsychotic actions and do not tackle cognitive impairments. Thus, a shift towards glutamate homeostasis recovery seems a promising strategy.

The Authors describe the title subject comprehensively and clearly. The only issue is that the Summary section refers only to the animal part of the studies described in the review. However, the manuscript equally reports also research on humans.

Author Response

We have changed the summary paragraph to refer to both human and animal studies, as suggested by the Reviewer. 

Reviewer 3 Report

The manuscript ‘Beyond NMDA receptors: homeostasis at the glutamate tripartite synapse and its contributions to cognitive dysfunction in schizophrenia’ by  Hagar Bauminger and Inna Gaisler-Salomon is interesting and very carefully and thoroughly prepared. However, I miss the reference to one review article, relevant to the topic discussed by the Authors, namely 'The tripartite glutamatergic synapse' by Ulyana Lalo et al. (PMID: 34433089).  Moreover, I have a comment about the descriptions of the table. They are located both above and below the tables. The information placed below should only be footnotes. 

Author Response

Thank you for your comments.

  1. We added the suggested review (ref 111) and refer to it on lines 299 (section on mGluRs) and 396 (EAATs).
  2. We removed the descriptions of the Tables from below and left only the footnotes.